Journal of
open psychology data

# Data from LEO 2018 – Living with Low Literacy

DATA PAPER

KLAUS BUDDEBERG 

KRISTIN SKOWRANEK 

GREGOR DUTZ 

ANKE GROTLÜSCHEN 

*Author affiliations can be found in the back matter of this article

]u[ ubiquity press

## ABSTRACT

The study "LEO 2018 – Living with Low Literacy" examines the reading and writing skills of adults aged between 18 and 64 years in Germany. It includes a literacy assessment and an extensive background questionnaire containing sociodemographic variables as well as information on literacy-related everyday practices and domain-specific basic skills (digital, financial, health-related, policy-related). The data was collected in 2018 as part of a household survey ($N$ = 7,192). The dataset is available for secondary use at the repository of *GESIS Leibniz Institute for the Social Sciences* as a Public Use File and as a Scientific Use File. The dataset offers a reuse potential for different research fields like financial literacy, health literacy, political literacy, digital literacy and with three variables about vulnerability even for psychological research questions.

**CORRESPONDING AUTHOR:**

**Klaus Buddeberg**

Universität Hamburg, DE

klaus.buddeberg@uni-hamburg.de

**KEYWORDS:**
Literacy; Basic Competences; LEO study; Adults; Germany

**TO CITE THIS ARTICLE:**

# (1) BACKGROUND

The data presented in this paper stems from the second round of the German literacy assessment called LEO 2018 – Living with Low Literacy (Buddeberg et al., 2021; Grotlüschen et al., 2023; Grotlüschen & Buddeberg, 2020). The original data is available in German, but was translated into English for the international research community (questionnaire, codebook, data set).

The LEO 2018 survey assessed the reading and writing skills of the German-speaking adult population aged between 18 and 64 years (Grotlüschen, Buddeberg, et al., 2020b).

The aim of the LEO 2018 survey is to investigate the extent of the phenomenon of low literacy skills among the German-speaking population and their participation in daily life practices, especially literacy related practices.

The survey reported about reading and writing skills in the classification system of the so-called Alpha-Levels which provide a differentiated scale for the lower levels of reading and writing proficiency in the German language (Grotlüschen, Nienkemper, & Duncker-Euringer, 2020). The Alpha-Levels are built on a theoretical framework in the context of the project LEA (Literalitätsentwicklung von Arbeitskräften/Literacy Development of the Workforce) (Grotlüschen et al., 2011) and are based on theoretically derived can-do descriptions with difficulty-determining characteristics and on the basis of theories of literacy acquisition. These theoretical assumptions were empirically confirmed in the first LEO survey (leo. – Level-One Survey), where they were subjected to a scaling test and finally used and validated for the first time in a population-representative way (Grotlüschen et al., 2012).

With some simplification Alpha-Level 1 can be assigned to mastering the level of letters, Alpha-Level 2 to mastering of the word level, and Alpha-Level 3 to mastering of the level of simple sentences. However, in order to accurately determine the difficulty of items, the length and usage of words must also be considered, as well as their phoneme structure (e.g., consonant clusters) (Grotlüschen & Riekmann, 2011, p. 65).

The range of low literacy (Alpha-Levels 1 to 3) roughly corresponds to level 1 and below in the *Programme for the International Assessment of Adult Competencies (PIAAC)* (Buddeberg et al., 2020).

The notion of literacy in LEO 2018 is threefold and includes (a) literacy as a measurable and scalable construct in the sense of a basic competence, but also (b) the use of reading and writing skills in everyday actions (literacy as a social practice), and finally (c) self-reported basic competences in different domains (digitalisation, health, finance, politics). Adults with low reading and writing skills are referred to as "low literate adults" (in German: gering literalisierte Erwachsene) instead of using terms like functional illiteracy. Especially people with skills at the Alpha-Levels 1 to 3 may experience various limitations in participation in daily life practices. The LEO 2018 survey considers their participation in various self-reported practices and self-reported basic skills in the areas of digitalization, health, political engagement and finances (Grotlüschen, Buddeberg, et al., 2020b). Additionally, it focuses on text related practices in the context of work, family and everyday life and asks about participation in continuing education, immigration and multilingualism (Grotlüschen et al., 2019).

Main results obtained with the data refer to the number (about 6.2 million) and proportion (12.1 percent) of low literate adults in Germany. Correlations exist between the level of literacy and formal education, age, migration and heritage language as well as socioeconomic and family background (Grotlüschen, Buddeberg, et al., 2020a). While differences between low literate and higher literate adults appear to be surprisingly small regarding employment or family status a higher degree of vulnerability exists regarding self-reported basic skills in different domains.

The survey data have been linked to the literacy-scale of PIAAC. Therefore, both studies and their scales can be related to each other (Buddeberg et al., 2020).

# (2) METHODS

## 2.1 STUDY DESIGN

The dataset of LEO 2018 is based on a random sample of German speaking adults. This means that the interviewees had to be capable to follow the interview in German. If the interviewers realized that an adult did not show sufficient oral command of German the respective interview was cancelled. During the interview the interviewees answered a standardized background questionnaire which was carried out as a *Computer Assisted Personal Interview (CAPI)*. All questions in the background questionnaire were read to the interviewees by the interviewers and for some questions of the assessment via audiofiles. In addition, every interviewee completed a reading and writing skills test (Paper and Pencil). After completing this test, respondents with a low score were handed an additional test with easier tasks (Figure 1). This procedure served to be able to differentiate the area of low literacy more precisely, because those who could be identified with weak results in the screening were to receive items tailored to them in the second step. The interviews were carried out by the survey institute Kantar Public, Munich (Grotlüschen et al., 2019; Bilger & Strauss, 2020).

The data were collected as a cross-sectional survey in face-to-face interviews. Every interview on average took about 49 to 60 minutes including the background

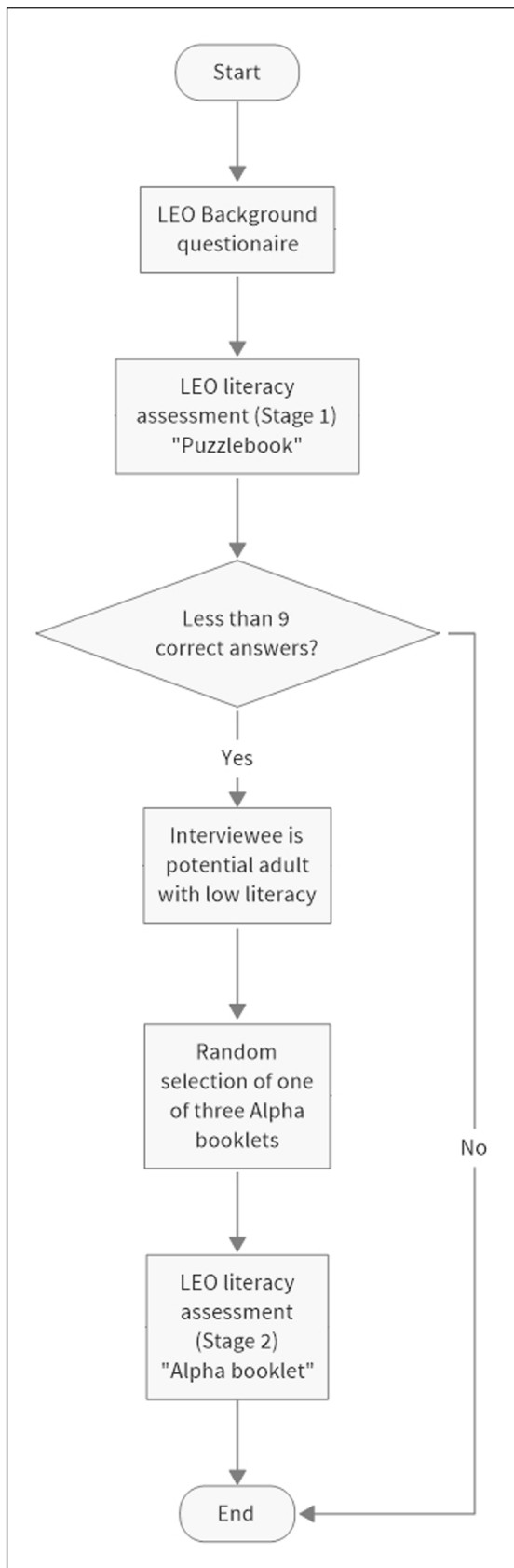

**Figure 1** Study Design of LEO 2018.

questionnaire and the reading and writing assessment. The interview time depended on the need for an extended literacy assessment (see chapter 2.5) (Bilger & Strauss, 2020, p. 92).

## 2.2 TIME OF DATA COLLECTION

The data was collected from March 1st 2018 until September 3rd 2018.

## 2.3 LOCATION OF DATA COLLECTION

The net sample covers the German speaking resident population aged 18 – 64 years living in private households in the Federal Republic of Germany (Bilger & Strauss, 2020, p. 90). The additional random sample covers the resident population aged 18 to 64 years living in private households in the Federal Republic of Germany with a "Hauptschulabschluss", which is equal to a general education certificate for leaving school after the 9th grade in Germany, or a lower or none school qualification (Bilger & Strauss, 2020, p. 91). Data was assessed in all German federal states, and the sampling procedure (described in 2.4) ensured adequate representation.

## 2.4 SAMPLING, SAMPLE AND DATA COLLECTION

The dataset of the LEO 2018 survey contains data from a net sample with 6,681 adults aged between 18 and 64 years and data from an additional sample with 511 adults aged between 18 and 64 years with a low or no school degree. The net sample contains data from the persons of the target populations who could be questioned (Diekmann, 2020, p. 377). The additional sample was necessary to generate enough test data from people with a low education level (Bilger & Strauss, 2020). Data was collected as CAPI.

The selection of the 6,681 respondents aged between 18 and 64 years in the net sample was made using a multi-layered three-step process that follows the transparency standards of the *German Business Association for Market and Social research (in German: Arbeitskreis Deutscher Markt- und Sozialforschungsinstitute e.V.; in short: ADM)* (Bilger & Strauss, 2020, p. 90).

During the first step the regional layering for the sample was determined. This first step was based on the 53,000 sample points (Häder, 2016) which split Germany up into comparable points nationwide whose structure corresponds to the distribution of private households. Out of these 53,000 sample points 1,300 sample points were selected by random sampling (Bilger & Strauss, 2020, p. 90).

In a second step, the households for the sample were selected using a random route. A random address whose household was not included in the survey was the starting point for a random walk (Häder, 2016). On the way of the random walk every third household was selected as part of the sample and it was checked whether persons living in these households met the criteria of the LEO 2018 survey. If there was no one living meeting the criteria, the household was counted as failure (Bilger & Strauss, 2020, pp. 90–91).

 

The target persons in a household were selected in a third step. Since every person that fitted the criteria of the survey living in one household should get the same chance to be part of the random sample, the selection was made with the help of the Kish-Selection-Grid (Bilger & Strauss, 2020, p. 91). The Kish-Selection-Grid provides that the interviewer first selects all household members that count as part of the target population. By randomly assigning digits printed on the questionnaire to the number of persons in the household, one person on the household is randomly selected for the interview (Diekmann, 2020, p. 384).

The additional random sample contains the data of 511 persons with a low level of education. The screening of this sample was based on a regular CAPI omnibus survey at Kantar Public. It includes individuals who had agreed to be interviewed again. Out of these persons the additional random sample was drawn with the characteristics school degree and age while care was taken to ensure that the age groups 18 through 34 years, 35 through 49 years and 50 through 64 years were distributed proportionally to the population (Bilger & Strauss, 2020, p. 91).

Participation in the interviews required a certain oral command of German. During the interview the interviewers read all questions from the background questionnaire to the interviewees (some items of the assessment included audio files) and entered the answers into the CAPI instrument. This measure made the interview neutral in terms of literacy because the interviewees did not need to read and write themselves. Therefore, there was no literacy related barrier for participation (Bilger & Strauss, 2020, p. 87).

Every interviewee got an incentive of 10 Euros which was announced at the beginning of the interview and was payed to the interviewees after finishing the tasks of the tests. The incentive was also payed when an interviewee decided to terminate the test. An exception to this rule was a termination of the test within the first two tasks (Bilger & Strauss, 2020, pp. 87–88).

Even if the dataset was sampled with great care there are some limitations in the data. Immigrants that do not speak German could not be included in the data as well as refugees in community accommodation. From the German speaking population disabled people who live in a home for handicapped people, inmates in prison and people older than 64 years could not be included in the sample (Grotlüschen, Buddeberg, et al., 2020b, p. 55).

## 2.5 MATERIALS/SURVEY INSTRUMENTS

The background questionnaire of the LEO 2018 survey was carried out as CAPI. During the CAPI the interviewers read the questions to the interviewees and put the answers back into the CAPI system. Because of this it was a low-threshold questionnaire even for people with low reading and writing skills and the method

can be described as literacy neutral (Bilger & Strauss, 2020, p. 87). By programming the CAPI instrument errors could be kept to a minimum by only allowing certain possible answers. The instrument also showed implausible answers (Bilger & Strauss, 2020, p. 94). Such contradictions could occur e.g. in the context of the household structure. First, the number of persons living in the household of the target person was asked, then the number of persons of certain age categories living in the household. Any inconsistencies that might arise at this point were indicated by the CAPI and an appropriate request was generated to resolve the inconsistency. The CAPI allowed using complex filters in the questionnaire (Bilger & Strauss, 2020, p. 87).

The background questionnaire gathered sociodemographic information about the interviewees. Additionally, there were questions about reading and writing practices and basic competences in the areas of digitalization, politics, health and finances. This information has not been tested but rely on self-reports. Questions about work, family, continuing education and migration were also included.

After finishing the background questionnaire, every interviewee received a so called "puzzlebook" with eleven tasks (each of them containing one or more test items) which relate to leisure time activities to avoid the impression of an exam situation. The tasks were carried out as paper and pencil. The results of the puzzlebook were transferred into the CAPI right after the interviewees finished it in presence of the target person. In order to keep this process of coding eleven answers as short as possible, the interviewers were instructed to score more harshly (i.e. as wrong) in cases of doubt. As all tasks were reviewed later in the coding process, any possible misjudgments could be corrected. This procedure ensured that, in case of doubt, a target person with low literacy skills was given another test booklet.

In total, eleven points (correct answers) could be reached in the puzzlebook. If a target person reached nine or more correct answers the assessment ended at this point. If this number of points was not reached, the interviewees received one of three alpha booklets which were selected randomly by the CAPI. These additional tasks were also carried out as paper and pencil. Low performing respondents thus received additional easy items in the booklets which were appropriate to their low reading and writing skills (Bilger & Strauss, 2020, p. 84).

The person's abilities measured on the basis of the assessment were plotted on a common continuous scale with the item difficulties using the *item response theory (IRT)* (Dutz & Hartig, 2020, p. 66). This scale was normalized in LEO 2010 to a mean of 50 and a standard deviation of 10. LEO 2018 and LEO 2010 were linked using common items and are therefore comparable. Certain sections of this scale mark levels, the so-called Alpha-Levels. The thresholds were determined theoretically

on the basis of difficulty-determining characteristics of certain items and were validated empirically (Hartig & Riekmann, 2012). For the scaling of the items the Rasch model for dichotomous response data was used. Estimates for person ability were calculated in the form of ten plausible values (Dutz & Hartig, 2020, p. 68). The Alpha-Levels 1 to 3 are a measure for differentiating the Level 1 in PIAAC or in the *International Adult Literacy Survey (IALS)* more detailed (Grotlüschen, Nienkemper & Duncker-Euringer 2020).

## 2.6 QUALITY CONTROL

Kantar Public used 367 interviewers which were trained especially for the LEO 2018 survey using the *train the trainer* method (Bilger & Strauss, 2020, p. 92). Based on this approach the research team at the University of Hamburg trained interviewers as regional multipliers using training material provided by the research team (Bilger & Strauss, 2020, p. 89).

Interviews which were not conducted correctly were excluded from the dataset (Bilger & Strauss, 2020, p. 93). Kantar Public used their self-developed similarity check to exclude fake survey data (Bilger & Strauss, 2020, p. 93).

To ensure quality control while scoring the test booklets various measures have been implemented. All personnel involved in the scoring of the test booklets were trained by the scientific staff of the University of Hamburg using a set of specified rules. These rules were also used in the previous survey, LEO 2010. To ensure comparability between LEO 2018 and 2010 about 300 test booklets from LEO 2010 were re-evaluated and results were compared to the previous evaluation showing strong consistency. The scoring of the data from 2010 was not changed during this procedure. Likewise, during the scoring process ten percent of all test booklets from LEO 2018 were evaluated by a second person to ensure consistency. Interrater reliability was controlled. During the process, which lasted several weeks, regular meetings were held to discuss cases of doubt, i.e. cases in which the editors were unsure whether to edit an answer as correct or not. Throughout the evaluation invalid entries like different handwritings or comments from the interviewers within one test booklet could be identified (Bilger & Strauss, 2020, p. 93).

## 2.7 DATA ANONYMIZATION AND ETHICAL ISSUES

The data collection was carried out by the Kantar Institute. Kantar is a member of the ADM and is certified with the standard of the European Society for Opinion and Market Research (ESOMAR). The data collection thus followed the ethical and quality standards defined there.

The data set of LEO 2018 was anonymized using the recommendations of the *Research Data Centre for Education (FDZ Bildung)* located at the *Leibniz Institute for Research and Information in Education (DIPF)* (www. forschungsdaten-bildung.de/get_files.php?action=get_ file&file=fdb-informiert-nr-3.pdf). Due to these recommendations, (a) individual values or categories of variables were coarsened, or (b) entire variables were coarsened, or (c) complete variables were deleted, depending on the need for anonymization.

In a first step of the anonymization the critical variables were identified. These were all data which could be assigned to specific persons like area information, detailed work information, education and especially foreign educational qualifications, country of birth, first language spoken in childhood, nationality, age, income and all open-ended variables.

The anonymization of the LEO 2018 data set followed the strategy of anonymization of the PIAAC data set (Rammstedt et al., 2016). Especially the coarsening of the countries of birth, nationality and the language of origin was based on the PIAAC anonymization. Due to this, countries were aggregated with similar countries if less than 50,000 inhabitants from those countries live in Germany. The categories were also used in the German Microcensus (Perry et al., 2017, pp. 19–20).

## 2.8 EXISTING USE OF DATA

The main publications resulting from the data from the LEO 2018 survey are Grotlüschen and Buddeberg (2020), Grotlüschen et al. (2023), Buddeberg, Dutz, Heilmann, et al. (2021), Buddeberg, Dutz, and Stammer (2021), Grotlüschen et al. (2019), Grotlüschen, Buddeberg, et al. (2020c), Grotlüschen, Nienkemper, and Duncker-Euringer (2020) and Smythe et al. (2021).

In addition to these publications there were papers published in journals and collections which are based on the LEO 2018 survey datasets. A list of these publications is available on the project blog (https://leo.blogs.uni-hamburg.de/publikationen/) and at the GESIS – Leibniz-Institute for the Social Sciences (https://search.gesis.org/research_data/ZA6266).

## (3) DATASET DESCRIPTION AND ACCESS

The dataset of the LEO 2018 survey includes 606 variables for 7,192 respondents from a random sample of the population aged 18–64 years residing in private households in Germany. The data is available as *Public Use File (PUF)* and as *Scientific Use File (SUF)*. The PUF can be accessed after registration on the website of the GESIS – Leibniz-Institute for the Social Sciences. The SUF can be obtained and processed by researchers after signing a data use agreement (contact via the PIAAC Research Data Center). It will be verified if (a) the applicant pursues scientific purposes, (b) belongs to a scientific institute

and (c) the planned research question can be addressed with the data. All other documents (e.g., questionnaires, codebooks, reports) are accessible without restrictions from the PIAAC Research Data Center website (www.gesis.org/en/piaac/rdc) as well as the GESIS – Leibniz-Institute for the Social Sciences.

Workshops for using the datasets have so far been offered once a year in online format. If there is interest and a sufficient number of participants, further tutorials can be arranged. Technical instructions (in German language) for data use are available at the project website: https://leo.blogs.uni-hamburg.de/wp-content/uploads/2022/11/221202-LEO-Workshop-R.pdf. Video material on data use is available on the project website: https://leo.blogs.uni-hamburg.de/einfuehrung-in-die-datennutzung-und-sekundaeranalysen-mit-dem-datensatz-von-leo-2018/ (permanent link).

### 3.1 REPOSITORY LOCATION
The datasets from the LEO 2018 survey are stored at the GESIS – Leibniz-Institute for the Social Sciences in Cologne, Germany. The PUF in German and English is available under DOI: https://doi.org/10.4232/1.13771. The SUF in German and English is available under DOI: https://doi.org/10.4232/1.13770.

Files, codebooks and the questionnaires are available through the GESIS – Leibniz-Institute for the Social Sciences in Germany and can be accessed through their website: https://search.gesis.org/research_data/ZA6266 (Public Use File) and https://search.gesis.org/research_data/ZA6265 (Scientific Use File).

### 3.2 OBJECT/FILE NAME
PUF: Grotlüschen, A., Buddeberg, K., Dutz, G., Heilmann, L. M., & Stammer, C. (2021). *LEO 2018 – Living with Low Literacy, Public Use File* (ZA6266; Version 1.0.0.). GESIS – Leibniz-Institute for the Social Sciences, Cologne. https://doi.org/10.4232/1.13771

The PUF is available in SPSS and Stata:

- ZA6266_v1-0-0.sav (German); ZA6266_v1_en.sav (English)
- ZA6266_v1-0-0.dta (German); ZA6266_v1_en.dta (English)

SUF: Grotlüschen, A., Buddeberg, K., Dutz, G., Heilmann, L. M., & Stammer, C. (2021). *LEO 2018 – Living with Low Literacy, Scientific Use File* (ZA6265; Version 1.0.0). GESIS – Leibniz-Institute for the Social Sciences, Cologne. https://doi.org/10.4232/1.13770

The SUF is available in SPSS and Stata:

- ZA6265_v1-0-0.sav (German); ZA6265_v1_en.sav (English)
- ZA6265_v1-0-0.dta (German); ZA6265_v1_en.dta (English)

The following documentation is available:

> Background questionnaire: ZA6265_fb.pdf (German); ZA6265_q.pdf (English)
> Codebook: ZA6265_cod.xlsx (German); ZA6265_cod_en.xlsx (English)
> User notes: ZA6265_Nutzungshinweise.pdf (German)
> Method report: ZA6265_mb.pdf (German)

### 3.3 DATA TYPE
Data represents primary data gathered in the context of the survey LEO 2018 – Living with low Literacy. Data includes survey data and scored performance data. The datasets in German and English are delivered for use with R, SPSS and Stata. The LEO project team recommends analyzing the data with the open source program R.

General instructions for analyzing the data like the background questionnaire, the usage notices and the method report are available as pdf. Detailed instructions for analyzing the data with R are provided in German on the project blog (https://leo.blogs.uni-hamburg.de/einfuehrung-in-die-datennutzung-und-sekundaeranalysen-mit-dem-datensatz-von-leo-2018) (permanent link).

### 3.4 LANGUAGE
The PUF, the SUF the Codebook and the questionnaire are available in English and German. The usage notices and the method report are available in German.

### 3.5 LICENSE AND LIMITS TO SHARING
The datasets from the LEO 2018 survey are available as PUF and SUF. In the PUF sensitive information was anonymized. The PUF is available at the GESIS – Leibniz-Institute for the Social Sciences and can be accessed after registration. The SUF contain full information and in compliance with data protection regulations (European Union General Data Protection Regulation [EU-GDPR]), is available for scientific research only and will be made available after signing a data use agreement (contact via the PIAAC Research Data Center).

### 3.6 PUBLICATION DATE
The dataset was published in July 2021 at the GESIS – Leibniz-Institute for the Social Sciences.

### 3.7 FAIR DATA/CODEBOOK
The PUF and the SUF in German and English conform to the FAIR Principles.

- Findable
  https://search.gesis.org/research_data/ZA6266 (Public Use File) and https://search.gesis.org/research_data/ZA6265 (Scientific Use File)
- Accessible

The PUF and the SUF in German and English are accessible through GESIS – Leibniz-Institute for the Social Sciences.

- Interoperable
  Datasets in German and English and documentation files in German and English are available in standard formats (R, SPSS, Stata, Word, Excel, PDF).
- Reusable
  For all datasets (PUF and SUF in German and in English), additional documents are available through the PIAAC Research Data Center webseite (e.g., questionnaire, codebook, manual for use with R, SPSS and STATA).

## (4) REUSE POTENTIAL

An interdisciplinary publication released in spring 2023 (Grotlüschen et al., 2023) has shown that the data are not only of interest for the field of adult and basic education research, but also for school and media pedagogy, sociology and social economics. The breadth of the questionnaire's content (digitization, finance, health, politics, work, family, migration) also makes it suitable for use by other academic disciplines such as migration research, political science, and health science.

The dataset offers variables for different research fields. Unlike most international surveys the dataset contains variables about *basic competencies*. In the field of *financial literacy* for example usually a wide range of competencies is assessed e.g. using the big-three-construct for financial literacy (Lusardi, 2015). The data provided in the LEO dataset however reports about very basic competencies, following the assumption that for adults facing a high risk of social vulnerability (e.g. poverty, precarious work) investment strategies or stock trading are far out of reach. For this group – and for research about this group – competencies in calculating budgets or comparison of prices are far more relevant. For research projects which focus on less privileged adults the data are a source of variables on finance related practices (transferring money, using online banking, being responsible for financial matters in the household, keeping written records of finances, searching for information to compare prices) and on self-reported specific competencies (preparing a tax declaration, selecting appropriate telephone or electricity providers, selecting an appropriate retirement provision, evaluate the benefits and risks of instalment purchases and online banking) (Buddeberg, 2020). These variables can be related to socioeconomic and sociodemographic factors as well as to the level of reading and writing skills. Due to the linking of the LEO scale with the PIAAC reading scale (see section 1) (Buddeberg et al., 2020) comparisons with the PIAAC survey might be performed. LEO 2018

contains differentiated data in the lower competence areas while PIAAC covers reading literacy from the low literacy range to the high literacy range (Buddeberg et al., 2020).

The data on digital competencies provided by the LEO dataset also do not imply information about high-level skills but like in the case of financial competencies more fundamental skills and practices. The possibility to correlate digital basic skills and digital practices to competence values on the literacy scale offers the opportunity to elaborate about digital skills and the level of social vulnerability represented by sociodemographic and socioeconomic variables and by the reading and writing skills. Variables about digital practices imply the use of telephone, desktop computer or mobile devices; need for assistance when using the internet, ways of digital writing like email, chat, social networks; attributes of digital writing like using emoticons or abbreviations; information search on the internet, use of audiovisual formats like online-tutorials or voicemail. Variables on digital basic competences in the dataset imply the use of online job portals, online housing portals, the competence to evaluate the credibility of text on the internet, the competence to distinguish between information and advertisement, and the competence to evaluate why internet companies have an interest in user data (Buddeberg & Grotlüschen, 2020). Again, due to the linking of the LEO and the PIAAC scale comparisons with PIAAC results can be drawn.

Data contains variables about reading and writing practices in the context of households and families. Therefore, from an economic perspective questions about intrafamilial time allocation might be examined. Intrafamilial processes regarding the transmission of literacy practices (e.g. reading to the children, visiting public libraries with the children or giving assistance preparing school exams) might link the perspectives of adult education and school education. Similar possibilities of researching basic competencies (instead of high-level competencies) exist for the fields of health (Heilmann, 2020) and politics (Dutz & Grotlüschen, 2020).

The data contain three specific variables related to vulnerability. They represent the general life satisfaction and the feeling of belonging to the community as well as satisfaction with the work-situation. These variables might be used for correlating them with different levels of reading and writing skills for psychological research. Since the data are representative of the German-speaking resident population (18-64 years), corresponding analyses are based on a solid database. It should be noted that the sample does not include persons without any German language skills and does not include persons living in shared accommodations, homes or prisons.

## SPECIAL COLLECTION

Collection: Data for Psychological Research in the Educational Field.

Editors: Sonja Bayer, Katarina Blask, Timo Gnambs, Malte Jansen, Débora Maehler, Alexia Meyermann, Claudia Neuendorf (alphabetic order).

## ACKNOWLEDGEMENTS

The data were generated with the advice of a scientific project advisory board:

Prof. Dr. Heike Solga, Wissenschaftszentrum Berlin

Prof. Dr. Helmut Bremer, Universität Duisburg-Essen

Prof. Dr. Dr. h. c. mult. Ingrid Gogolin, Universität Hamburg

Prof. Dr. Johannes Hartig, DIPF Frankfurt

Prof. Dr. Klaus Hurrelmann, Hertie School of Governance, Berlin

Prof. Dr. Carola Iller, Universität Hildesheim

Prof. Dr. Bernd Käpplinger, Justus-Liebig-Universität Gießen

Prof. Dr. Corinna Kleinert, LIfBi, Universität Bamberg

Prof. Dr. Nele McElvany, Technische Universität Dortmund

Prof. Dr. Beatrice Rammstedt, GESIS Mannheim

Prof. Dr. Doris Schaeffer, Universität Bielefeld

Prof. Dr. Josef Schrader, Deutsches Institut für Erwachsenenbildung, Bonn

## FUNDING INFORMATION

The project, in the context of which the described data were collected, was funded by the German Federal Ministry of Education and Research (Bundesministerium für Bildung und Forschung, BMBF) under the funding code W146600.

## COMPETING INTERESTS

The authors have no competing interests to declare.

## AUTHOR CONTRIBUTIONS

Contributors to this publication are the authors Klaus Buddeberg, Gregor Dutz, Anke Grotlüschen and Kristin Skowranek. Former members of the project staff who also contributed to the data sets are Lisanne Heilmann, Christopher Stammer, Caroline Euringer, Franziska Bonna and Anna Heimböckel. Frauke Bilger and Alexandra Strauß were responsible for the field work by Kantar. The project was supported by a scientific advisory board (see acknowledgements).

## AUTHOR AFFILIATIONS

**Klaus Buddeberg** orcid.org/0000-0002-0416-8092
Universität Hamburg, DE

**Kristin Skowranek** orcid.org/0000-0002-1018-7097
Universität Hamburg, DE

**Gregor Dutz** orcid.org/0000-0002-0115-3768
Universität Hamburg, DE

**Anke Grotlüschen** orcid.org/0000-0003-3072-1741
Universität Hamburg, DE

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

## PEER REVIEW COMMENTS

*Journal of Open Psychology Data* has blind peer review, which is unblinded upon article acceptance. The editorial history of this article can be downloaded here:

- **PR File 1.** Peer Review History. DOI: https://doi.org/10.5334/jopd.91.pr1

**TO CITE THIS ARTICLE:**

