## [Peer Review History. · Journal of Open Psychology Data]

Dear editors of the JOPD and dear reviewers,

We want to thank you for your reading our paper and providing your valuable questions, comments and recommendations.

As a result, we have made numerous changes in the structure of the manuscript, e.g. moving certain passages to other sections. Therefore, we did not use the revision mode for displaying the changes we have made but answered your points in the table below.

We hope that you agree with the revision we have made and thank you for having another look into the article.

Best wishes,

Corresponding author

Particular important points:

Review	Answer
1) I found some parts of the manuscript redundant or misplaced in some sections. For example, the information on the scaling procedure on Page 4 would be better included in Section 2.5 where the instruments are described. Moreover, the information on the sample size and timing of data collection seems partly redundant to the information presented in subsequent sections. Therefore, I recommend revising the manuscript to identify and remove redundancies and better align the presented information with the subsections used in the manuscript. Both reviewers had more specific suggestions similar to mine.	Thank you for that advice. We have checked for redundancies. The specific recommendations of the two reviews were also very helpful.
2) I strongly recommend describing the instruments in greater detail. Particularly, what information was assessed in the background questionnaire? Was this limited to sociodemographic information or were additional scales administered? How many items were administered in the test? I believe that would help better understand the collected information.	We described the background questionnaire in more detail. We also added information about the number of problems that respondents had to solve in the puzzle booklet and in the alpha booklet.
3) In Section 2.8, please cite the respective publications and put the references in the reference section. In my opinion, it is not essential to list the publications currently given under "some relevant international publications"	We have followed this recommendation and omitted the small reference list at

because you refer to the two websites listing all publications resulting from the data set. If you want to explicitly mention these publications, I would like to ask you to follow the second reviewer's recommendation and briefly mention the main topic/results of the cited publication.	this stage of the manuscript.
4) Your data is not available under an open data license which is also emphasized in Section 3.6. This is no major problem because the data can still be accessed after registration. However, I hope you could clarify the criteria for users to be granted access to the scientific use files. Can anybody signing a data usage contract access the data or are there specific conditions/limitations on data usage?	We have added detailed information about the SUF application process.
5) Section 4 on the reuse potential is rather short and very unspecific. I strongly recommend expanding this section to outline more concrete research questions that might be addressed with the data. Please take a look at other articles published in the special issue as a guideline (https://openpsychologydata.metajnl.com/collections/data-for-psychological-research).	We have added a number of more concrete examples of possible re-use of the data.

Reviewer A

Review	Answer
However, the authors state in the paper (section 3.6 License) that both datasets are not deposited under an open license and further provided no information on the adherence to ethical standards or consent forms (except for the information on the procedure of anonymization).	In section 2.7, we have added information on the quality measures and ethical standards applied by the institute that carried out the fieldwork.
(1) Information about the instruments used for assessment is missing. Please include information on the background questionnaire (e.g. number of items, constructs assessed, example questions, and if applicable scales used and reliability) as well as the items used for assessing the reading and writing skills (e.g. number of items, constructs assessed, scales used, reliability, example items).	Thank you for this comment. We have added information on the aspects you mentioned.
(2) I would recommend restructuring the methods section. The structure provided by the template is not very well implemented. For example, information on the study design or the sample are scattered over the different subsections (e.g. in the subsections "Time of data collection" and "Location of data collection"). This makes it harder for the reader to gather all relevant information from the section. In particular, I would recommend to:	Both reviewers provided valuable advice on reorganizing parts of the manuscript. We have followed these recommendations and hope that the information will be easier to find in the document.
- Move the first paragraph in the methods section to the study design subsection	We moved this paragraph
- Move the first paragraph in the study design subsection to the sampling subsection	We moved this paragraph
- Remove the information on the time of data collection in the study design subsection, as it is redundant and not relevant in this subsection	We removed this information
- Move the third paragraph in the study design subsection to the materials subsection, as it is rather information about the instruments and data analysis than on the study design	We moved this paragraph
- Move the information on the sample in the time subsection to the sampling subsection	We moved this information
(3) A table or graph would be helpful to visualize the study design and instruments used in the study.	We decided to use a flowchart to visualize the study design.
(4) Please explain the abbreviations used in the graph provided for the sampling design. Also consider designing it more clearly (e.g., less text) and with a larger font size.	We decided to delete the graph and use a flowchart instead, as this was recommended in the other review.
(5) Please review your use and explanations of abbreviations and make them consistent. Explain all abbreviations (also PISA, ICILS, ...) when you first use them (e.g., by adding the abbreviation in parenthesis: "...Scientific Use File (SUF)...") and use the abbreviation consequently afterwards.	We reviewed all our abbreviations in the document and made them consistent.
(6) I recommend avoiding (frequent) repetitions of the same information or even same parts of sentences. For	We removed the sentence "assessed

example, the sentence "...assessed the reading and writing skills of the German-speaking adult population aged between 18 and 64 years (Grotlüschen, Buddeberg, et al., 2020b)" is written twice in short distance in the background section. In general, some information (e.g., on the sample or the age of the participants) is mentioned repeatedly within the paper but not always relevant in the respective section.	the literacy skills of the German-speaking adult population aged 18-64" in the second place and rewrote the second sentence.
(7) Consider checking the punctuation.	We have checked this point.
(1) Background section	
- I would not refer to large-scale assessments as "events", as some are conducted repeatedly and not just once	As we decided to delete the first paragraph, this aspect no longer appears.
- "Adult skills" should be written in lower case letters when not referring to a name of an assessment	We have decided to delete the first paragraph. Therefore, "adult skills" only appears in the reference list.
- "These theoretical assumptions were empirically confirmed in the preliminary work for the studies": which studies are you referring to?	We have added information on these studies.
- "Alpha Level" is sometimes written with and without hyphen, and once even only as "Alpha"	We have standardized the term Alpha-Level or Alpha-Levels with a hyphen.
- "PIAAC level 1 and below": I would recommend explaining this for readers who are not acquainted with the PIAAC levels	The phrase no longer appears in the document.
- "Main results refer...": rather "Main results obtained with the data refer..."	We have changed the sentence accordingly.
- "Correlations exist with...": please state as well explicitly the variable/construct that correlates with the mentioned factors (e.g., low literacy or proportion of low literacy)	We have clarified this point by describing that there are correlations between levels of literacy and other factors.
(2) Methods section	
- "i.e. who could be assumed to have low literacy skills": I would recommend removing this part of the sentence or splitting the whole sentence in two	We removed that part of the sentence.
- "Data was collected as Computer Assisted Personal Interviews (CAPI). The data were collected as a cross-sectional survey in face-to-face interviews...": repetition	After moving a few paragraphs, this repetition was gone.
- Consider explaining terms shortly, e.g., net sample, sample points, random route, Kish-Selection-Grid	We explained the terms net sample and kish selection grid. The terms sample points and random route were already explained in the text. We checked if we

	could explain these terms in more detail.
- “The additional random sample contains the data of 511 persons from the lower levels of education”: consider rephrasing the last part, e.g. in “with a low level of education”	We rephrased the last part of the sentence
- “If a certain number of points was not reached in the puzzlebook”: What do you mean exactly with points? How many points?	We clarified this passage by adding the threshold of at least 9 out of 11 correct answers.
- “asks after special interview criteria”: I’m not sure if that is correct, consider rephrasing, e.g., “asks according to special...”	We rephrased this sentence.
(3) Dataset description section	
- “Both files together with the codebook and the questionnaire are available through the GESIS Data Archive in Germany and can be accesses through their website...”: it should rather be “...and can be accessed through their website...”	We adjusted the word accessed.
- “The files of the Scientific Use File are named after the same system”: I’m also here not sure whether this is correct. I would rather suggest “...follow the same naming scheme/system”	We rephrased this sentence.
- “This File does not contain all the variables in cause of anonymization of the data.”: I’m also here not sure if this is correct. I would rather write “This file does not contain all variables due to the anonymization of the data.”	We rephrased this sentence.
- “The Codebook provides all the necessary information which variables are accessible through the Public Use File.”: I recommend including “...about which...”	We rephrased this sentence
- Some closing parenthesis are missing in the subsection FAIR data/Codebook (e.g., after Scientific Use File)	We adjusted this error.
(4) Reuse potential section	
- “It should be noted that the sample does not include persons without any language skills“: What do you mean with “without any language skills”? Low literacy or no German language skills?	We had omitted the word German. We have corrected this error.
- This section is a bit short.	We have added a number of more concrete examples of how the data can be reused.

Reviewer B

Review	Answer
Abstract. The abstract mentions that a data use workshop can be offered. This is not further addressed in the main manuscript. I also wondered whether workshops or training materials could be provided on a long-term basis (e.g., as videos). Eventually, the authors should comment on how long workshop offers remain. Furthermore, including a sentence on the reuse potential would be valuable.	Thank you for this point. We have added a section on workshops in section 3.
(1) Background	
- The first paragraph is a general introduction to large-scale assessment and is not specific to the data presented. I'm also hesitant with statements, such as that LSA is the dominant approach in educational research, that need proper backup. The authors might consider beginning directly with what is currently the third and fourth paragraphs.	We thank you for this recommendation and have decided to delete the first paragraph entirely.
- I wondered whether specificities of the German language (e.g., transparent orthography) should be briefly summarized. This might be important for data users who seek to use this data for international comparisons. This could be done in the context of introducing the Alpha levels.	We have reorganized the whole section you refer to and added the information that the alpha levels have only been developed for assessments in the German language.
- The last sentence of paragraph 5 states that "theoretical assumptions were empirically confirmed in the preliminary work for the studies." A brief summary of (selected) evidence supporting the assumptions would be helpful.	We have reorganized this section and added information about the studies.
- Paragraph 8 reflects on terminology development between LEO 2012 and LEO 2018. I recommend briefly defining how specific terms are used in LEO 2018 instead of sketching out the terminological development.	Thank you for this suggestion. In fact, it is far more important for the reader to be informed about the terms currently in use. We have changed this passage.
- Last paragraph "The survey data have been linked to the literacy scale of PIAAC. Therefore, both studies and their scales can be related to each other." This important feature should be briefly described in section (4) on the reuse potential.	Thank you for this valuable advice. We have added a number of more specific examples of possible re-use of the data. In these examples we have emphasized the point you made.
2.1 Study design	
- The section describes details about the sample and the measurements, which I would have expected in other sections (i.e., 2.4 sample, 2.5. materials/measures). What is missing is information about the administration of the	We have restructured this section based on recommendations from the

background questionnaire, the "puzzle book", and the routing to an Alpha booklet. This information is spread across Figure 1 and section 2.5 on materials.	other review. We hope that the information provided in this section on study design is more coherent.
- A reference to Figure 1 needs to be included. The readability of the figure could also be improved by increasing the figure or the font. At least for me, the figure could be more intuitive. A flow chart could be better suited to visualize the study design.	Thank you for recommending a flowchart. We have decided to adopt the idea and remove the original figure. We have also added a reference to the figure.
2.3 Location of data collection	
- Just for clarification: Data was assessed in all German states, and the sampling procedure (described in 2.4) ensured adequate representation.	We added the sentence.
- The authors state that participants required basic German language skills. How was this baseline level defined and determined? It might improve readability if this information is added to 2.1 (study design) or 2.4 (sampling).	We specified this point by adding more detailed information in section 2.1.
- The authors write that the interviewer read the interview questions. However, Figure 1 denotes that the interviewer activated audio. The authors should clarify the mode. I also expected this information to appear in another section, like 2.4. (data collection)	We clarified this point by adding information about audio files. Moreover, we moved the passage to section 2.4.
- At the end of the paragraph, the authors conclude that the sample is representative based on participants' language abilities. I cannot follow this reasoning and would have expected this conclusion after describing the sampling procedure.	We deleted this passage
2.5 Materials / Survey instruments	
- I need clarification on the statement that the CAPI programming only allowed for possible answers, followed by the information that implausible answers were shown. Can the authors elaborate and provide examples?	We have added an example to illustrate where certain inconsistencies might occur and how the CAPI was used to generate additional questions to clarify the inconsistency.
- In the second paragraph, the authors describe that the results of the puzzle book task were transferred into the CAPI by the interviewer. How time-intensive was this process? What did the participants do in the meantime?	We've added the number of responses the interviewers had to code.
2.6 Quality control	
- Second paragraph: What are the in-house control procedures? Readers are likely not familiar with those. I also	We decided to delete this information

wondered to which effect standardized postcards or validation instruments were sent out. What exactly was done in this example? Also, how was it determined whether the interviews were conducted correctly? What were the criteria?	
- Third paragraph "To ensure comparability between LEO 2018 and 2010, about 300 test booklets from LEO 2010 were re-evaluated, and results were compared to the previous evaluation showing strong consistency.": Do the authors imply that the scoring of the previous study, LEO 2012, was retroactively changed? If not, they might consider rewriting this sentence.	We added a sentence to clarify that the scoring of the first survey was not changed during the process.
- The authors also state that two individuals coded 10% of the test booklets. Did the authors determine interrater reliabilities at some point? How were "cases of doubt" determined? Was it up to the coder's confidence, or were systematic criteria applied?	We have added information about controlling for interrater reliability and added a sentence to clarify 'cases of doubt'.
2.7 Data anonymization and ethical issues	
- The authors should clarify if they mean the DIPF Leibniz Institute for Research and Information in Education or the Research Data Centre for Education (FDZ Bildung), located at DIPF. I guess they mean the latter.	We clarified this point.
- Second paragraph: What is the variable "newspaper"?	We deleted this example and left the reference to "all open-ended variables".
- The authors should include information on ethical issues. Did the study adhere to ethical standards? How were participants asked for their consent? This is particularly important, as the study focused on adults with low literacy who likely need help understanding long and complex text.	Information on quality measures and ethical standards applied by the institute that carried out the fieldwork has been added to section 2.7.
2.8. Existing use of data.	
- As a reader, I prefer a summary of the research questions and main results of the work listed here. The single publications might be listed in an appendix or the references to improve readability.	We followed your recommendation
(3) Dataset description and access	
- How is the approval process to gain access to the Scientific Use File from the LEO 2018 project management?	We added detailed information regarding this question.
- It might be stated that LEO 2018 includes survey data and (scored) performance data in section 3.3 Data type.	We followed your recommendation and added the sentence: "Data includes survey data and scored performance data."

3.4 Format names and versions	
- The presented information does not seem to fit the section heading.	We moved the information of this section to section 3.3.
- Is the provided link a permanent link? To what degree do the usage notices and the project blog overlap?	We added the information that we offer permanent links. We also added information that the documents provided by GESIS and on the project blog are not identical.
(4) Reuse potential	
The author's suggestions for reuse are broad and unspecific. They might consider adding a few more specific suggestions for reuse. Some might include a reference to the PIAAC data.	We have added a number of more concrete examples of how the data can be reused.
General points.	
- There are various abbreviations throughout the manuscript that should be written out the first time when mentioned (e.g., FDZ GESIS in the abstract; PIAAC, PISA, etc. in (1) Background; CAPI in (2) Methods). Although they are common in LSA contexts, the authors should regard less knowledgeable readers.	We checked all our abbreviations in the document, wrote them out for the first time and made them consistent.
- Cross-references within the paper should be specific (e.g., "see section xyz" instead of "see above").	We checked all cross-references in the document, clarified them and made them consistent.
- The manuscript reads alright, but I still recommend language proofreading to check for typos, commas, wording, unnecessary redundancies, etc.	We checked the document for the aspects you mentioned.